# A plasma-based protein signature association with all-cause mortality

Natalia Koziar[1☉], Anthony D. Whetton[1,2☉], Nophar Geifman [3☉]*

**1** School of Biosciences, Faculty of Health and Medical Sciences, University of Surrey, Guildford, United Kingdom, **2** Veterinary Health Innovation Engine, School of Veterinary Medicine, University of Surrey, Guildford, United Kingdom, **3** School of Health Sciences, Faculty of Health and Medical Sciences, University of Surrey, Guildford, United Kingdom

☉ These authors contributed equally to this work.
* ng0023@surrey.ac.uk

## Abstract

Circulating plasma proteins play key roles in measuring and reflecting states of disease and health. Developments in protein metrology allow for over 2,900 proteins to be quantified in a single sample. In major epidemiological studies, this allows for profound insights into protein expression in liquid biopsies, mortality, and morbidity. Here, we have investigated the relationship between peripheral blood protein profiles and non-accident all-cause mortality within 5- and 10-year timeframes, using data on 38,150 participants from the UK Biobank. Adjusting for lifestyle and health covariates, we identified 392 proteins associated with an increase in risk for death within 5-years, and 377 associated with an increase in risk within 10-years. Proteomic signatures of cause-specific mortality (cardiovascular, cancer, all-other causes) were also identified, with 19 proteins found to overlap across those. Using logistic regression modelling, we constructed a parsimonious predictive protein panel for each respective all-cause mortality timeframe, including markers such as adrenomedullin, SERPINA1 and PLAUR. When compared to models inclusive of standalone traditional risk criteria, such as demographic and lifestyle factors, models utilising the protein panels modestly improve prediction for 5 and 10-year mortality (from AUC 0.49–0.57 to AUC of 0.62–0.68). Our results demonstrate the potential of the plasma proteome in risk stratification for all-cause mortality.

## Introduction

In any given population, a relatively small proportion of patients will account for a disproportionately large fraction of health care costs. Earlier identification of patients in need of intervention is viewed as a means of reducing costs to health care providers; in England alone, approximately 70% of all health expenditure is attributed to just 30% of the population [1]. Risk stratification is acknowledged as crucial for the

**Data availability statement:** The data used in this study is available to any researchers upon application to the UK Biobank. For full details please see: https://www.ukbiobank.ac.uk/use-our-data/apply-for-access/.

**Funding:** The author(s) received no specific funding for this work.

**Competing interests:** The authors have declared that no competing interests exist.

identification of individuals at most risk of adverse health outcomes, enabling targeted interventions that improve patient prognosis and optimise resource allocation. Projecting future mortality, even when a primary disease diagnosis is not present, is therefore a means of potentially increasing patient and health economic benefits via a personalised medicine approach.

Circulating plasma proteins are in part surrogates for the physiological and pathological processes occurring in various tissues and organs, offering insight into the influences and interactions of environmental factors, genetics, and medications [2]. Several proteins have already been recognised as specific diagnostic assays, such as N-terminal pro-B-type natriuretic peptide (NT-proBNP) for heart failure and alanine aminotransferase (ALT) for hepatocellular carcinomas [3,4]. However, the clinical utility of broad-spectrum proteomics to simultaneously assess risk for specific diseases and also across a wide range of morbidities remains to be established. Several studies have employed multiple plasma protein analytes in machine learning models for the prediction of disease onset in the UK Biobank [5–7]. Although, for the majority of conditions, proteomic-based approaches only marginally improve prediction power when compared to the standalone usage of traditional risk assessment tools, such as lifestyle parameters and clinical biomarkers.

Leveraging data on 2,917 plasma protein relative quantification measurements in 38,150 individuals from the United Kingdom Biobank (UKB), [8] We aimed to determine whether risk of all-cause mortality is reflected by a small number of circulating protein markers. Furthermore, we assess whether such a parsimonious proteomic signature can be informative of, contribute with, or outperform conventional risk assessment criteria when predicting all-cause mortality within 5 or 10 years of sample collection (see study design in Supporting Fig 1). A data-driven approach to assessing the proteome in individuals, along with data on time of death has enabled a focused assessment of markers that are potentially measurable in a primary care setting for assessing mortality risk for patient benefit.

## Results

Using the UKB dataset, we identified volunteers who had suffered non-accidental death within either 5 or 10 years from the time of initial assessment. Study participants ranged between 39 and 70 years old, and included 769 individuals who died from disease-related causes within 5 years of recruitment, and 2,100 who died within 10 years (Table 1). For full exclusion and inclusion criteria, see Methods and Supplementary Tables 1 and 2 in S1 File. With the exception of the number of diagnosed chronic conditions, no significant difference in demographics where found between the 5-year and 10-year mortality groups.

Electronic health data linkage was used to categorise and determine the frequencies of specific-cause mortality in this cohort (Table 2 and Supplementary Table 3 in S1 File). Ischemic heart disease was the leading cause of disease-specific mortality, accounting for 10.8% of deaths, with all cardiovascular-related mortality attributed to approximately 21% of all deaths. Lung cancer was the second leading

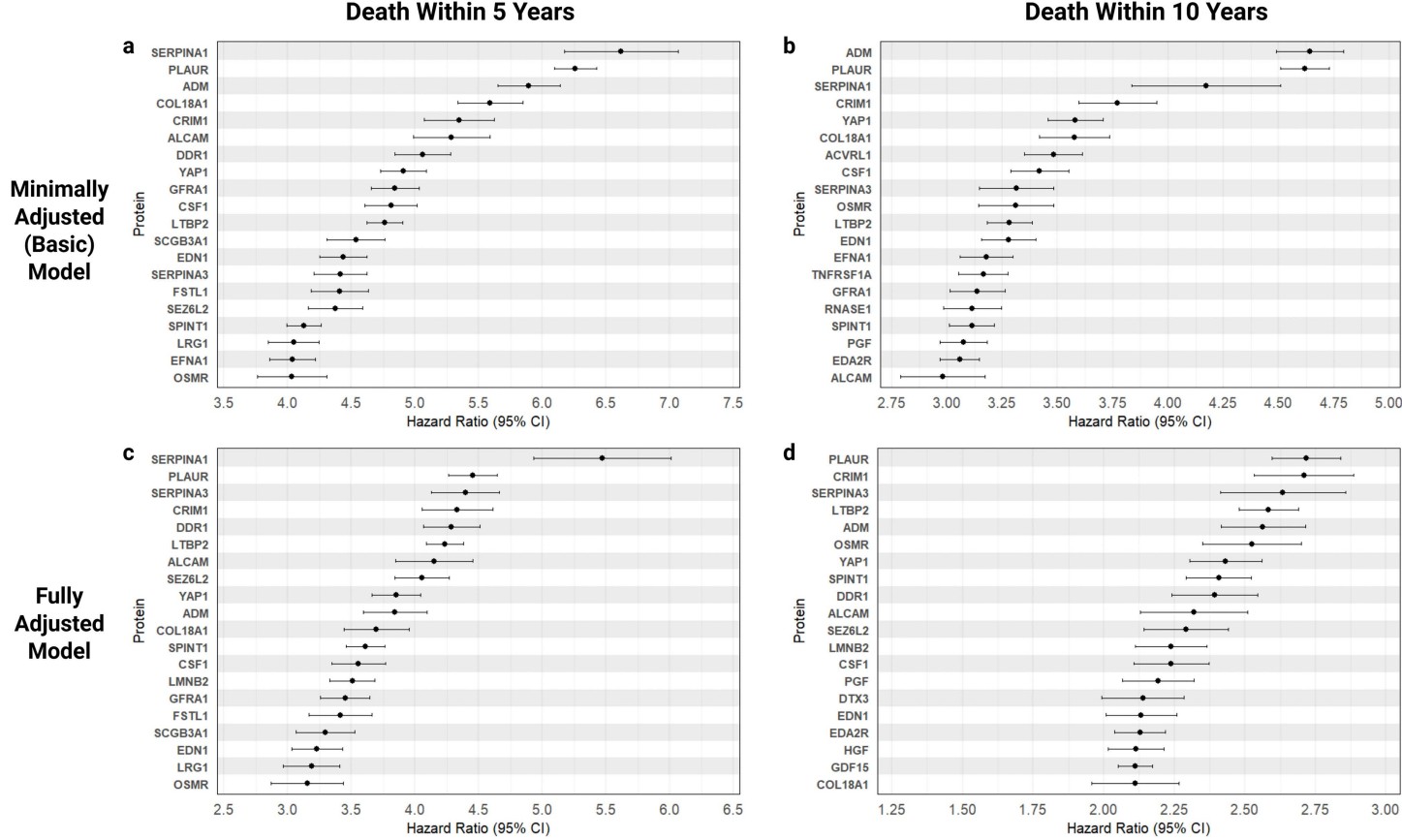

**Fig 1. Effect sizes on all-cause mortality.** Statistically significant (p < 1.71 × 10−5) proteins with the largest effect sizes on all-cause mortality within 5 (n = 769) or 10 years (n = 2,100). Associations were tested using Cox proportional hazards models, with the minimally adjusted model being adjusted for age, sex and BMI only, and the fully adjusted model adjusted for age, sex, BMI, chronic condition count, alcohol, smoking, physical activity, education and Townsend deprivation index. **a**, 20 proteins most indicative of risk (HR > 1) for all-cause mortality within 5 years in the minimally adjusted model. **b**, 20 proteins most indicative of risk for mortality within 10 years in the minimally adjusted model. **c**, 20 proteins most indicative of risk for mortality within 5 years in the fully adjusted model. **d**, 20 proteins most indicative of risk for mortality within 10 years in the fully adjusted model.

disease-specific cause of death (8.4%), with cancer-related mortality overall being the leading cause of death in the cohort, accounting for almost half (47.8%) of all deaths within 10 years of sample collection.

## Circulating proteins association with all-cause mortality

Proteins significantly associated with an increased risk of all-cause mortality were determined using Cox proportional hazard (CPH) modelling (Fig 1. and Supplementary Tables 4-7 in S1 File). We derived two approaches, a minimally adjusted model which included the basic demographic factors of age, sex and BMI, and a fully adjusted model which further included chronic condition count, alcohol intake frequency, smoking status, physical activity, education and Townsend deprivation index.

Out of 2,917 proteins, 558 were found to be statistically significant (p < 1.71 × 10−5) for 5-year onset of all-cause mortality in the minimally adjusted model, increasing to 690 proteins for 10-year onset. The majority of proteins produced a hazard ratio (HR) above 1, indicating an association with an increase in risk for all-cause mortality. This consisted of 535 and 647 proteins for mortality within 5 and 10 years, respectively. Serpin family A member 1 (SERPINA1), plasminogen activator urokinase receptor

**Table 1. Summary of phenotypic data used in this analysis from the 38,150 individuals who met inclusion criteria, participants who had missing entries or chose "not to answer" were excluded.**

| | Proteomic Cohort | Deceased Within 5 Years | Deceased Within 10 Years | *P Value |
|---|---|---|---|---|
| **Number of Participants** | 38,150 | 769 | 2,100 | |
| **Age (years)** | | | | 0.968 |
| Range | 39-70 | 40-70 | 40-70 | |
| Mean | 56.7 | 61.8 | 61.8 | |
| **BMI (weight/height²)** | | | | 0.377 |
| Range | 14.5-54.9 | 15.9 - 53.9 | 15.9 - 69.0 | |
| Mean | 27.3 | 28.0 | 28.2 | |
| **Sex** | | | | 0.179 |
| Female | 20,528 | 487 | 1272 | |
| Male | 17,622 | 282 | 828 | |
| **Chronic Condition Count** | | | | $2.21 \times 10^{-7}$ |
| Range | 0-21 | 0-13 | 0-16 | |
| Mean | 1.9 | 3.5 | 4.1 | |
| **Education** | | | | 0.670 |
| College or University degree | 12,808 | 185 | 498 | |
| A levels/AS levels or equivalent | 4,320 | 55 | 194 | |
| O levels/GCSEs or equivalent | 8,091 | 153 | 420 | |
| CSEs or equivalent | 2,042 | 30 | 90 | |
| NVQ or HND or HNC or equivalent | 2,518 | 67 | 172 | |
| Other professional qualifications | 2,019 | 53 | 131 | |
| None of the above | 6,352 | 226 | 595 | |
| **Townsend Deprivation Index** | | | | 0.687 |
| Range | −6.3 - 10.2 | −6.2 - 9.9 | −6.2 - 10.4 | |
| Mean | −1.3 | −0.7 | −0.7 | |
| **Alcohol Intake Frequency** | | | | 0.963 |
| Daily/ almost daily | 7,904 | 172 | 497 | |
| 3-4 times per week | 8,716 | 144 | 397 | |
| 1-2 times per week | 9,982 | 184 | 490 | |
| 1-3 times per month | 4,131 | 67 | 189 | |
| Special occasions only | 4,311 | 110 | 279 | |
| Never | 3,106 | 92 | 248 | |
| **Smoking Status** | | | | 0.624 |
| Never | 20,764 | 278 | 799 | |
| Previous | 13,402 | 345 | 923 | |
| Current | 3,984 | 146 | 378 | |
| **Moderate Physical Activity (days/week)** | | | | 0.901 |
| 0 | 4,967 | 146 | 405 | |
| 1 | 3,055 | 50 | 138 | |
| 2 | 5,529 | 98 | 257 | |
| 3 | 5,756 | 100 | 291 | |
| 4 | 3,750 | 67 | 184 | |
| 5 | 5,713 | 111 | 280 | |
| 6 | 2,118 | 42 | 114 | |
| 7 | 7,262 | 155 | 431 | |

*Association between phenotype and all-cause mortality within 5 versus 10 years. For continuous data (age, BMI, chronic condition count, deprivation index and physical activity), p-value was calculated using two sample t-test, for categorical data (sex, education, alcohol intake and smoking status), p-value was calculated using chi-squared test of independence.

**Table 2. Top 20 categorised causes of death in 2,100 individuals with proteomic data that have died within 10 years of blood sample collection.**

| Category | Included ICD10 codes | Number of Deaths | Percentage of Deaths |
|---|---|---|---|
| Ischemic heart disease | I214, I219, I249, I250, I251, I254, I258 | 226 | 10.76 |
| Lung cancer | C341, C349, C37, C383, C384, C399 | 178 | 8.48 |
| Motor neuron disease | G122 | 133 | 6.33 |
| Colorectal cancer | C180, C181, C182, C187, C189, C19, C20, C210 | 107 | 5.09 |
| Breast cancer | C509 | 100 | 4.76 |
| Hematopoietic and lymphoid tissue cancer | C819, C829, C833, C837, C838, C844, C845, C851, C859, C880, C900, C901, C910, C911, C917, C920, C921, C97, D469, D474 | 91 | 4.33 |
| Cerebrovascular disease | I607, I609, I614, I619, I620, I629, I632, I634, I639, I64, I679, I690, I691 | 82 | 3.90 |
| Chronic obstructive pulmonary disease | J440, J441, J449 | 70 | 3.33 |
| Pancreatic cancer | C250, C259 | 63 | 3.00 |
| Interstitial pulmonary diseases | J840, J841, J849 | 61 | 2.90 |
| Prostate cancer | C61 | 57 | 2.71 |
| Brain cancer | C710, C712, C714, C719 | 57 | 2.71 |
| Gynaecological cancer | C519, C541, C55, C56, C570, C578, C579 | 53 | 2.52 |
| Oesophageal cancer | C159 | 51 | 2.43 |
| Dementia | F019, F03 | 51 | 2.43 |
| Other forms of heart disease | I309, I330, I350, I359, I38, I420, I421, I422, I429, I442, I461, I472, I48, I489, I499, I515, I516, I517, I518 | 48 | 2.29 |
| Cancer with ill-defined, secondary and unspecified sites | C786, C80, C800, C809 | 46 | 2.19 |
| Other diseases of the nervous system | G039, G061, G062, G119, G231, G310, G318, G319, G35, G713, G903, G931, G934, G959, G961, G98 | 46 | 2.19 |
| Soft tissue cancer | C450, C459, C480, C482, C491, C492, C493, C495, C499 | 44 | 2.09 |

(PLAUR) and adrenomedullin (ADM) were the three proteins most indicative of risk within the minimally adjusted models, with hazard ratios of 6.62, 6.26 and 5.89 for all-cause mortality within 5 years, respectively. These decreased marginally for 10-year onset but still demonstrated a significant increase in risk (HR > 4) in individuals with elevated levels of these markers.

In the fully adjusted model, the number of statistically significant proteins fell to 401 for all-cause mortality within 5-year onset, and 385 proteins for 10-year onset. Most proteins remained associated with an increase in risk (HR > 1); these included 392 and 377 proteins for mortality within 5 and 10 years, respectively. Compared to the minimally adjusted model, the effect sizes of hazard ratios were notably smaller, and protein markers associated with the largest increases in risk changed depending on the event window. For 5-year all-cause mortality, SERPINA1, PLAUR and serpin family A member 3 (SERPINA3) were the proteins most indicative of risk, with respective hazard ratios of 5.47, 4.45 and 4.33. Effect sizes decreased for 10-year onset but still illustrated a considerable increase in risk (HR > 2). Interestingly, this was the only model to not feature SERPINA1, however PLAUR and SERPINA3 remained as two of the top 3 markers most indicative of risk, alongside cysteine rich transmembrane BMP regulator 1 (CRIM1).

## Protein association with disease-specific mortality

To determine whether there was a difference in proteomic signatures for disease-specific mortality, we employed cox proportional hazard modelling to identify the proteins with the largest effect sizes on cardiovascular, cancer, and all other related mortalities within 10 years of blood sample collection. (Fig 2 and Supplementary Table 8 in S1 File).

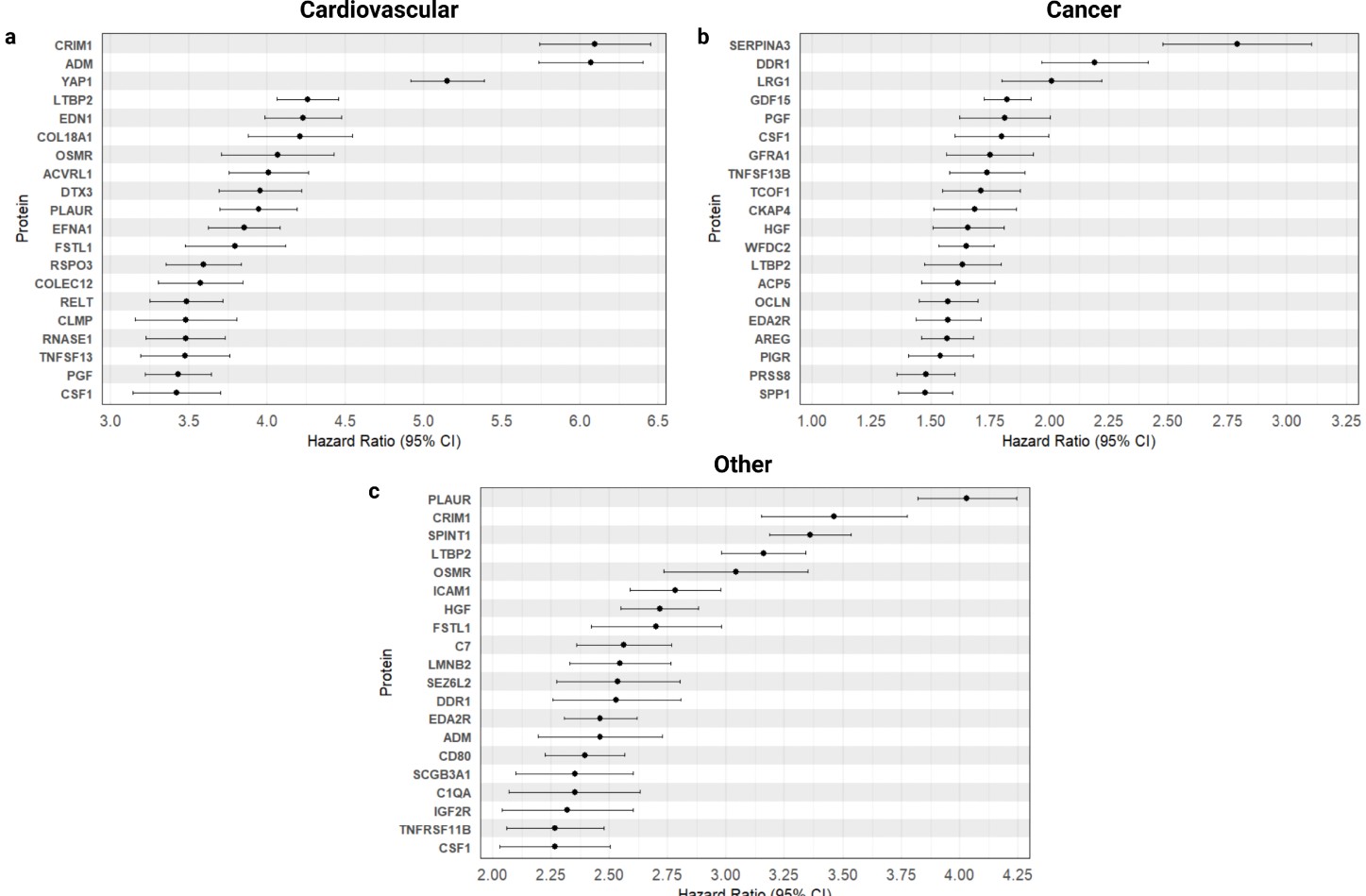

**Fig 2. Effect sizes on disease-specific mortality.** Statistically significant (p < 1.71 × 10 − 5) proteins with the largest effect sizes on disease-specific mortality within 10 years using the fully adjusted model. **a**, 20 proteins most indicative of risk (HR > 1) for cardiovascular-specific mortality within 10 years (n = 441). **b**, 20 proteins most indicative of risk for cancer-specific mortality within 10 years (n = 1003). **c**, 20 proteins most indicative of risk for all other mortality within 10 years (n = 656).

For cardiovascular-specific cause of deaths, a total of 312 proteins were discovered to be significantly associated. CRIM1 and ADM exhibited the highest hazard ratios (HR > 6), followed by YAP1, LTBP2, and EDN1 (with hazard ratios of 5.15, 4.26 and 4.23, respectively). For mortality attributed to cancer, 28 proteins were statistically significant, with SER-PINA3 showing the largest effect (2.79), alongside DDR1, LRG1, and GDF15 as the top contributors. (2.19, 2.01 and 1.82, respectively). For mortality from all other causes, 139 proteins were statistically significant. PLAUR had the highest hazard ratio of 4.03, followed by CRIM1, SPINT1 and LTBP2 (with hazard ratios of 3.46, 3.36 and 3.16, respectively). A total of 19 significantly associated proteins were shared across all three cause-specific mortality groups, with several (CRIM1, ADM, SERPINA3, DDR1, PLAUR and LTBP2) among the top proteins significantly associated with all-cause mortality; however, these HRs were lower than the respective cause-specific HRs for several proteins (e.g., CRIM1 and PLAUR).

## Cumulative risk prediction

Following the initial association analysis, logistic regression modelling was used to calculate cumulative area under the curve (AUC) values for proteins found to be significantly associated with increase in risk (HR > 1) for all-cause mortality within the fully adjusted CPH model (Fig 3 and Supplementary Tables 11 and 12 in S1 File). Iteratively adding one protein at a time, going by their HR ranked order from highest to lowest, the models provide an estimate of the discriminatory ability of protein signatures to distinguish between survivors and those who died within the specified timeframe. To combat the issue of class imbalance (as the number of deaths within 5 and 10 years represented only 2.0% and 5.5% of the cohort, respectively) we employed propensity score matching, where the number of deceased individuals (769 for 5 years and 2,100 for 10 years) was matched to an equal number of surviving participants based on age, sex, deprivation index, and number of diagnosed chronic conditions.

For all-cause mortality within 5 years, a baseline of approximately 0.55 AUC was observed with the first protein (SERPINA1), increasing to approximately 0.66 AUC with the addition of the next five proteins: PLAUR, SERPINA3, CRIM1, discoidin domain receptor tyrosine kinase 1 (DDR1) and latent transforming growth factor beta binding protein 2 (LTBP2). Addition of the following several proteins either decreased or made no significant difference to the cumulative AUC, until the incorporation of TNF receptor superfamily member 1A (TNFRSF1A), growth differentiation factor 15 (GDF15) and interleukin 1 receptor type 1 (IL1R1), which increased cumulative AUC to approximately 0.68.

For 10-year risk, a baseline of approximately 0.61 AUC was provided by the first protein (PLAUR). Compared to the shorter event window of 5 years, the increase in cumulative AUC was more gradual. The following nine proteins: CRIM1, SERPINA3, LTBP2, ADM, oncostatin M receptor (OSMR), yes-associated protein 1 (YAP1), serine peptidase inhibitor Kunitz type 1 (SPINT1), DDR1 and activated leukocyte cell adhesion molecule (ALCAM) resulted in a cumulative AUC of approximately 0.63, increasing to an uppermost value of 0.64 with the addition of further proteins.

## Model comparison and evaluation

We next constructed a predictive panel for each event window using the proteins that contributed to an increase in cumulative AUC (Fig 3). This included six proteins for 5-year risk (SERPINA1, PLAUR, SERPINA3, CRIM1, DDR1 and LTBP2) and ten proteins for 10-year risk (PLAUR, CRIM1, SERPINA3, LTBP2, ADM, OSMR, YAP1, SPINT1, DDRI and ALCAM). Using logistic regression modelling, the prediction power of the protein panels was measured and compared to that of traditional risk criteria alone, or in combination with (Fig 4).

Models that included only basic demographic information (age, sex and BMI) showed the lowest predictive performance, with AUC values close to 0.50, indicating near-random prediction. The addition of lifestyle factors only marginally improved AUC values and performance metrics, increasing the AUC by approximately 10% to 0.554 and 0.567 for 5-year and 10-year mortality, respectively. A longer event window improved specificity in demographic and lifestyle models, however, decreased sensitivity (0.457 to 0.456 and 0.535 to 0. 492 for the model including age, sex, BMI, and for the model including age, sex, BMI + lifestyle factors, respectively), indicating a decreased ability to identify individuals at risk when using phenotypic and lifestyle criteria (Fig 4).

UK Biobank also included data on blood biochemistry inclusive of the quantification of 30 standard clinical biomarkers, of which 9 and 22 were found to be significantly associated with all-cause mortality within 5 and 10 years, respectively (Supplementary Tables 9 and 10 in S1 File). We next considered the effect of adding these into consideration of mortality prediction. Briefly, those found to be statistically significant, such as cystatin C and C reactive protein, marginally improved AUC by approximately 12% and 8% for death within 5 and 10 years, respectively, when compared to the Age, Sex, BMI + Lifestyle model. However, the optimal AUC was observed in models utilising selected predictive protein panels, with the protein markers alone (Proteins Only model) improving AUC by approximately 33% when compared to the Age, Sex, BMI model for 5-year risk, and 15% for 10-year risk. Addition of demographic and lifestyle factors to the protein models further improved prediction and performance metrics, with the highest AUC (0.676) and sensitivity (0.609) values being

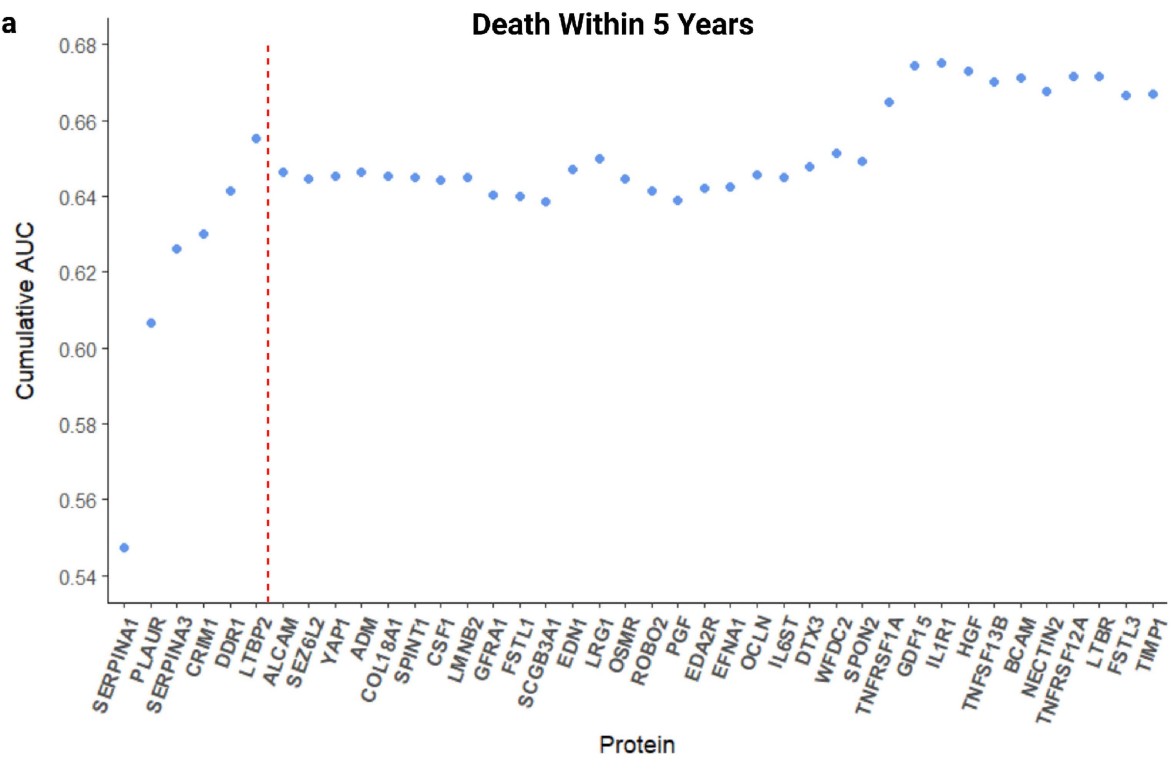

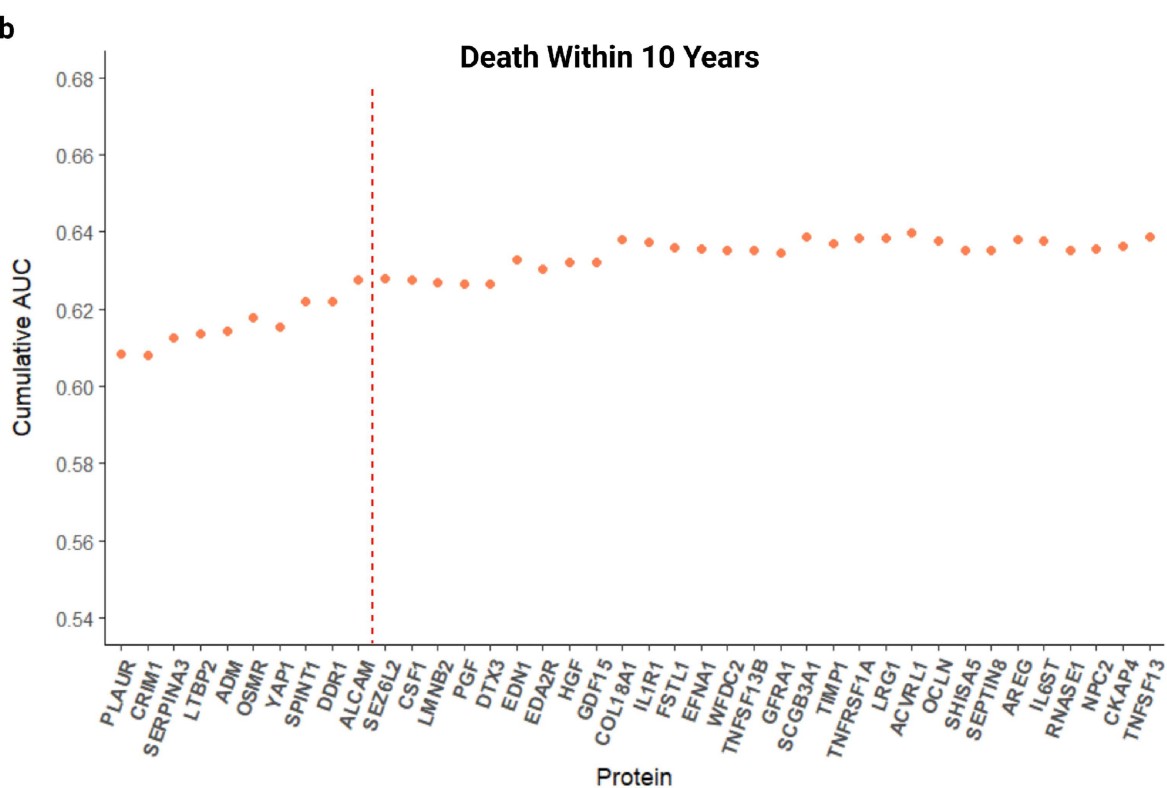

**Fig 3. Cumulative area under the curve (AUC) of 40 proteins with the largest hazard ratios from fully adjusted cox proportional hazards models.** A logistic regression model was run for each protein significantly associated with an increase in risk (hazard ratio > 1) for all-cause mortality within **a**, 5-year onset (n = 392) and **b**, 10-year onset (n = 377). A red, dashed line illustrates the protein selection cut-off for the panel.

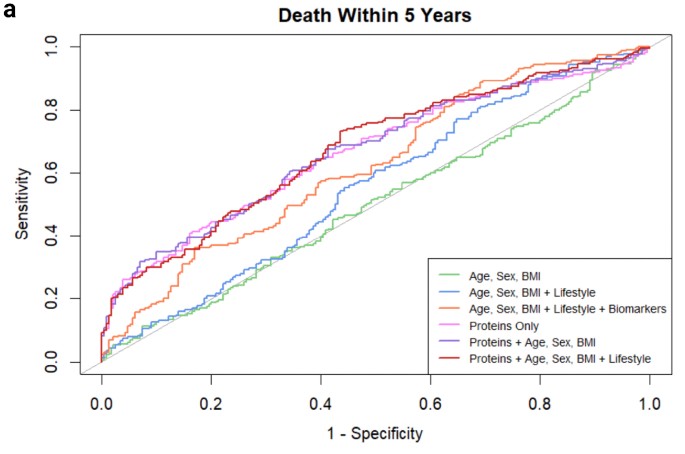

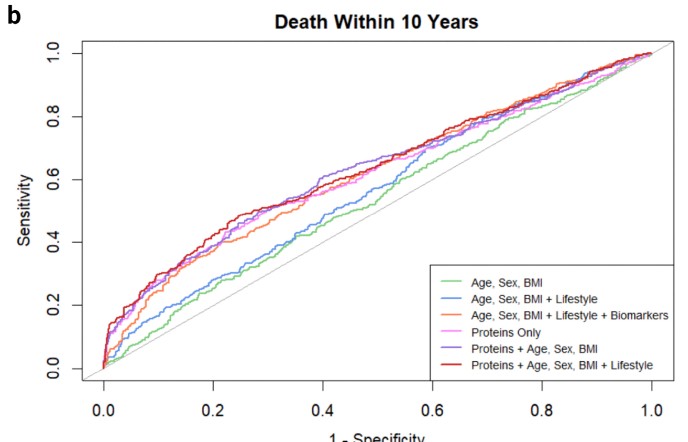

**c**

| Model | Death Within (Years) | AUC | Sensitivity | Specificity | Precision |
|---|---|---|---|---|---|
| Age, Sex, BMI | 5 | 0.497 | 0.457 | 0.570 | 0.515 |
| | 10 | 0.537 | 0.456 | 0.595 | 0.530 |
| Age, Sex, BMI + Lifestyle | 5 | 0.554 | 0.535 | 0.565 | 0.552 |
| | 10 | 0.567 | 0.492 | 0.587 | 0.544 |
| Biomarkers, Age, Sex, BMI + Lifestyle | 5 | 0.623 | 0.570 | 0.604 | 0.590 |
| | 10 | 0.615 | 0.548 | 0.622 | 0.592 |
| Proteins Only | 5 | 0.665 | 0.600 | 0.635 | 0.622 |
| | 10 | 0.616 | 0.548 | 0.619 | 0.590 |
| Proteins + Age, Sex, BMI | 5 | 0.670 | **0.609** | 0.635 | 0.625 |
| | 10 | 0.627 | 0.549 | 0.638 | 0.603 |
| Proteins + Age, Sex, BMI + Lifestyle | 5 | **0.676** | 0.604 | 0.635 | 0.623 |
| | 10 | 0.633 | 0.544 | 0.630 | 0.595 |

**Fig 4. Comparison of model performance for predicting all-cause mortality for 5 and 10 years.** "Proteins" refers to the predictive protein panel for the respective timeframe. "Lifestyle" covariates include the following: smoking status, alcohol intake frequency, chronic condition count, physical activity, Townsend deprivation index and education. "Biomarkers" consists of the UK Biobank measured biochemistry biomarkers which were found to be significantly associated with the respective timeframe of death. **a**, ROC curves for models predicting all-cause mortality within 5 years. **b**, ROC curves for models predicting all-cause mortality within 10 years. **c**, table comparing performance metrics of each model and timeframe.

observed in the 5-year risk Proteins + Age, Sex, BMI + Lifestyle model, and the 5-year risk Protein + Age, Sex, BMI model, respectively. However, this improvement was less noticeable for 10-year onset, suggesting a lesser impact of demographic and lifestyle factors in a longer prediction period.

## Discussion

Death is a multifactorial health outcome, influenced by a complex interplay of genetic, environmental, and behavioural factors, making it inherently difficult to predict. Nevertheless, there are known risk factors such as insufficient physical activity, abnormal body mass index and alcohol consumption that are associated with increased mortality in the general population [9–11]. Here, we examine associations between the proteome and the risk of death, investigating whether a shared, non-cause-specific mortality signature is reflected by a selection of circulating proteins, and whether these can enhance the prediction of all-cause mortality outcomes within 5 years or 10 years, beyond the traditional risk criteria of demographic and lifestyle-based risk factors. Our analysis found 392 and 377 proteins associated with an increase in risk for all-cause

mortality within 5 and 10 years, respectively, after adjusting for phenotypic, lifestyle, and health covariates. Following this, models incorporating a panel of just ten or fewer of the most predictive proteins for the corresponding timeframe, outperformed models utilising only demographic and lifestyle information, improving on AUC and performance metrics. Additionally, models incorporating the predictive protein panels surpassed routinely used blood biochemistry markers, though this was more notable for 5-year risk. However, the predictive discrimination ability of models utilising the protein panel was still relatively low (AUC of 0.62–0.68), prompting further investigation into the clinical utility of using proteomics for all-cause mortality risk stratification.

When examining the biological roles of the most predictive proteins identified, it becomes clear they are involved in a range of key processes, particularly signal transduction, cell differentiation and proliferation, and inflammation. While the majority of these proteins have been previously associated with mortality and poor prognosis in certain conditions, this is the first study to combine them into signature panels for all-cause mortality prediction. We found that the strongest predictors differed for 5-year and 10-year onset of mortality, potentially indicating that certain proteins are more reflective of more immediate, acute health risks, while others reflect more gradual, chronic processes that affect long-term survival. For example, high expression of SERPINA1, also known as alpha1-antitrypsin, was identified to be a strong predictor of 5-year risk of all-cause mortality. SERPINA1 is a protease inhibitor, whose main function is to protect tissues from the activity of enzymes, such as neutrophil elastase, that are released during inflammation [12]. Genetic variants of SERPINA1 have been linked to increased risk for conditions such as large artery atherosclerotic stroke, and high expression of the protein has been associated with poor prognosis in several cancers, including lung and colorectal [13–15]. ADM, a potent vasodilator with various functions such as tissue repair and anti-inflammatory effects, has been linked to disease severity and poorer prognosis in patients with conditions such as pulmonary arterial hypertension and left ventricular hypertrophy [16–18].

Five proteins (PLAUR, SERPINA3, CRIM1, DDR1 and LTBP2) were found to be key predictors of all-cause mortality regardless of timeframe, possibly suggesting key roles in the biological processes underlying both short-term and longer-term mortality risk. For example, PLAUR encodes the urokinase-type plasminogen activator receptor (uPAR), responsible for mediating a variety of functions such as signal transduction, plasminogen activation, and extracellular matrix degradation [19,20]. Over-expression of PLAUR within activation of the plasminogen system, has been correlated with poor survival and prognosis in several cancers, in addition to being predictive of adverse cardiovascular events [21–23]. Similar associations can be seen with SERPINA3, an acute-phase protein secreted into circulation during acute and chronic inflammation, and whose main function is inhibition of proteases such as neutrophil cathepsin G and pancreatic chymotrypsin [24]. Elevated levels of SERPINA3 are associated with lower survival rates in several malignant melanomas and carcinomas, where the protein is shown to promote cell migration and invasion [25,26]. Overexpression of SERPINA3 has also been demonstrated to contribute to the pathogenesis of neurodegenerative and cardiovascular disease, suggesting potential as a possible prognostic marker [27,28]. LTBP2 is an extracellular matrix protein with multiple functions, including cell adhesion and indirect regulation of transforming growth factor beta (TGFβ) activity [29]. While observed to have both tumour suppressing and tumour promoting functions, up-regulation of LTBP2 has been associated with poorer patient outcomes in certain cancers [30,31]. Additionally, it has been demonstrated that LTBP2 is a strong predictor of adverse cardiovascular events, including all-cause and pulmonary death [32–34].

The study design and the available data have several strengths, including a large prospective cohort sample, application of a high-throughput proteomics platform, a long follow-up period with links to health-records, and strict control for potential confounding factors to better isolate the direct relationship between the identified proteins and mortality. There are, however, some limitations. First, it should be noted that the identified protein signature does not necessarily infer causality for earlier death, and testing for the potential causal role of individual proteins was not possible. Furthermore, while a signature comprising of few proteins is favourable for clinical utility and translation to clinical decision taking, it may not be sufficient enough to capture the full spectrum of the genetic, environmental, and lifestyle interactions that contribute towards mortality risk.

A clinical predictive model is recommended to produce an AUC of 0.7 or greater, and whilst our models approach this value, the target of predicting an event 5 or 10 years with death caused by multifarious factors is a difficult proposition. Similar proteomic studies, notably Gadd D et al. (2024) and Smith A et al. (2025) [5,35], have achieved the recommended AUC values with the incorporation of substantially larger protein panels. This, however, makes biological interpretation and validation more challenging due to the difficulty of discerning which proteins are the major contributors behind the signature. Further, panels comprising of 100s of proteins are not practical for clinical testing. In contrast, our study focuses specifically on all-cause mortality, excluding accidental deaths, and leverages a broader proteomic panel to systematically identify a parsimonious set of proteins most strongly associated with death. Although the predictive improvements over traditional risk factors are modest, our approach balances predictive performance with interpretability, providing insight into the proteomic contributions to mortality that are distinct from multi-disease prediction networks.

Due to the unique nature of the UK Biobank, our study presently lacks an external validation, limiting the generalisability of our findings; further, as a result of the vast majority of UK Biobank participants being of European ancestry, the predictive power of our protein panel is yet to be established for other ethnic groups. Finally, while self-reported health and lifestyle habits are arguably less reliable, further work is required to assess the cost-effectiveness and net clinical benefit of protein measurements in mortality risk stratification against the easier and more inexpensive measurement of traditional risk criteria, albeit in our study we have found them to be similar to tossing a coin [36].

In conclusion, our analysis demonstrates that a shared, non-specific signature of mortality risk can be captured by a small set of proteins, while effect sizes differ by cause. This protein signature can improve on prediction value for all-cause mortality up to 5 and 10 years before death, when compared to basic demographic and lifestyle factors, as well as routinely used clinical biomarkers. The improvement in prediction power is, however, limited, and further research is required to evaluate the clinical utility of using a plasma protein panel in mortality risk stratification. Nonetheless, our data demonstrates that a scorecard reflecting protein levels can be derived to provide a quantifiable measure for evaluating associations between protein signatures and long-term mortality outcomes.

## Methods

### UK Biobank Study Cohort

Data was extracted from the UK Biobank database, a large-scale prospective study which recruited around half a million participants aged 40–69 years at baseline. Participants were enrolled from the 13th of March 2006 to the 21st of July 2010, visiting one of the 22 recruitment centres across the UK for an initial assessment. All participants provided written consent. Proteomic analysis was conducted on a subset of blood samples collected at recruitment, consisting of 46,673 randomly selected participants at baseline visit, and 6,385 individuals at baseline selected by the UKB-PPP consortium. At the time of data collection, 40 participants had withdrawn consent, resulting in a dataset of 53,018 individuals and 2,923 protein measurements. Data was first accessed 1st of December 2023. Authors did not have access to any identifiable information for participants.

### Proteomics

Circulating plasma proteins were identified using Olink Explore across four panels (Cardiometabolic I, Inflammation I, Neurology I, Oncology I), measuring 2,923 unique proteins. Details on sample preparation, processing, and quality control have been previously described [8]. 11,613 individuals had > 10% missingness across all protein measurements and were therefore excluded from analysis, alongside six proteins (CST, CTSS, GLIPR1, NPM1, PCOLCE, TACSTD2) that had > 10% missingness across all samples. Remaining missing protein values were imputed using half of the lowest value present in the dataset for each respective protein.

## Electronic health data linkage

Death records including cause and date were obtained through linkage to national death registries. 104 Participants whose cause of death was missing, or non-disease related were excluded from analysis. Causes of death defined as non-disease related were those relating to injury and other external causes, for which the respective ICD10 codes can be seen in Supplementary Table 1 in S1 File. Disease incident data was obtained from linkage to National Health Service records and included diagnosis in either primary or secondary care. Diagnoses were coded according to the International Classification of Disease version 10 (ICD-10), and relevant ICD10 codes were used to calculate chronic condition count. Diseases and the respective ICD10 codes used are available in Supplementary Table 2 in S1 File. Diseases were defined as chronic conditions based on already published literature investigating multimorbidity in the UK Biobank [37,38].

## Phenotype data in UK Biobank

Phenotypic information was obtained from the initial assessment and questionnaire each participant completed upon their visit to the recruitment centre. 3,151 individuals that had missing entries or chose "prefer not to answer" were excluded from analysis. Lifestyle covariates used included age, sex, BMI (weight in kilograms, divided by height in meters squared), chronic condition count (range from 0 to 21), education (1 = College or University degree, 2 = A levels/AS levels or equivalent, 3 = O levels/GCSEs or equivalent, 4 = CSEs or equivalent, 5 = NVQ or HND or HNC or equivalent, 6 = other professional qualifications, −7 = none of the above), alcohol intake frequency (1 = daily or almost daily, 2 = three or four times a week, 3 = once or twice a week, 4 = one to three times a month, 5 = special occasions only, 6 = never), smoking status (0 = never, 1 = previous, 2 = current), number of days per week of moderate physical activity (range from 0 to 7), and Townsend deprivation index (range from −6–10, with higher numbers correlating to higher levels of deprivation). Phenotypic data for the 38,150 individuals used in analysis can be seen in Table 1.

## Statistical analysis

Cox proportional hazards models were run for each protein using the 'survival' package (version 3–7.0) in R (version 4.2.2) with death specified as the outcome. Time to event was calculated in days, from date of initial assessment (and proteomic assessment blood sample collection) to date of death. Cut-off time was 1825 and 3650 days for 5 and 10 years, respectively. The basic model was adjusted for age, sex, and BMI only, while the fully adjusted model was additionally adjusted for education, chronic condition count, alcohol frequency, smoking status, levels of deprivation, and physical activity. A Bonferroni-adjusted P value threshold of $(0.05/2917) = 1.71 \times 10^{-5}$ was applied to correct for multiple testing. Summary statistics for each model are included in Supplementary Tables 4-8 in S1 File.

Statistically significant proteins from the fully adjusted model that were indicative of risk (hazard ratio > 1) for all-cause mortality were then ranked according to effect size and implemented in a logistic regression model using the glm() function in base R.

Using propensity score matching, individuals deceased within 5 and 10 years were matched to an equal number (770 and 2,100 for 5 and 10 years, respectively) of surviving individuals based on age, sex, Townsend deprivation index and number of diagnosed chronic conditions, using nearest neighbour matching from the "MatchIt" package (version 4.5.5). Data was then randomly split into training (70%) and test (30%) sets, before calculating cumulative AUC for each protein with the use of the 'pROC' package (version 1.18.5). Complete results can be seen in Supplementary Tables 11 and 12 in S1 File. The same dataset constructed with propensity score matching was then used to train and test the demographic, lifestyle and protein prediction models (see Fig 4) using logistic regression with k-fold cross validation (k = 10) with the 'caret' package (version 7.0−1).

## Supporting information

**S1 Fig. Study design summary for a plasma proteomic signature for all-cause mortality prediction (n = 38,150).** Cox proportional hazards (PH) models were used to measure associations between individual proteins and all-cause mortality within 5 and 10 years from the time of sample collection. Proteins that had associations of $p < 1.71 \times 10^{-5}$ (Bonferroni-adjusted threshold) in the fully adjusted (age, sex, BMI and lifestyle factors) were retained. Next, propensity score matching was used to match an equal number of survivors to the deceased participants for each event window based on age, sex, deprivation index and chronic condition count. Logistic regression was used to model the predictive value of each retained protein, from which a prediction panel was derived from based on contribution to cumulative AUC. Logistic regression modelling was then repeated with k-fold cross validation (k = 10) to evaluate performance of models incorporating the protein panel against those using clinical biomarkers and traditional risk factors. (TIF)

**S1 File. Supplementary Tables 1–12.** (XLSX)

## Acknowledgments

This research has been conducted using the UK Biobank Resource under Application Number: 83988. We thank the UK Biobank volunteers for their participation.

## Author contributions

**Conceptualization:** Natalia Koziar, Anthony D. Whetton, Nophar Geifman.

**Data curation:** Natalia Koziar.

**Formal analysis:** Natalia Koziar.

**Investigation:** Natalia Koziar.

**Methodology:** Natalia Koziar, Anthony D. Whetton, Nophar Geifman.

**Supervision:** Nophar Geifman.

**Validation:** Natalia Koziar, Anthony D. Whetton.

**Visualization:** Natalia Koziar.

**Writing – original draft:** Natalia Koziar.

**Writing – review & editing:** Anthony D. Whetton, Nophar Geifman.

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
