## [Decision Letter · Decision Letter 0]

7 Aug 2025

Dear Dr. Geifman,

We look forward to receiving your revised manuscript.

Kind regards,

Pavel Strnad

Academic Editor

PLOS ONE

Journal Requirements:

Reviewers' comments:

Reviewer's Responses to Questions

**Comments to the Author**

1. Is the manuscript technically sound, and do the data support the conclusions?

Reviewer #1: Partly

Reviewer #2: Yes

2. Has the statistical analysis been performed appropriately and rigorously?

Reviewer #1: Yes

Reviewer #2: Yes

3. Have the authors made all data underlying the findings in their manuscript fully available?

Reviewer #1: Yes

Reviewer #2: Yes

4. Is the manuscript presented in an intelligible fashion and written in standard English?

Reviewer #1: Yes

Reviewer #2: Yes

Reviewer #1: Comments to the authors

The authors analyse >2,900 plasma proteins in a large prospective UK Biobank cohort (n = 38,150) and propose 6- and 10-protein panels for predicting all-cause mortality at 5 and 10 years (best AUC ≈ 0.66). While the dataset is highly valuable and the topic is of importance, the current version of the manuscript contains significant conceptual and methodological flaws that must be addressed before the results can be properly evaluated. The manuscript could make a valuable contribution if it is revised to address cause-of-death heterogeneity and position its findings within the context of recent large-scale proteomics work.

Major points:

Outcome definition

The study treats all-cause mortality as a unified endpoint, however this aggregates biologically diverse causes of death like cardiovascular, cancer, respiratory, infection or neurological death. Without cause-specific breakdown or analyses, the current protein panel is difficult to interpret mechanistically or to apply clinically. For example, proteins associated with infection-related mortality may be diluted when pooled with proteins associated cancer deaths and vice versa.

A table summarizing leading causes of death by ICD-10 code, with a breakdown in numbers and percentages and corresponding stratified hazard models is needed to demonstrate the true scope and relevance of the findings. Biomarker panels for mortality have limited utility if they cannot clarify why a patient is at risk. For clinical use risk stratification linked to potential actions like cardiovascular prevention or cancer screening is of relevance. The authors should justify the focus on undifferentiated all-cause-mortality, or reframe their analysis around clinically distinct outcomes.

Relationship to prior work

Recent UK Biobank studies have already examined protein-based disease prediction across multiple outcomes:

• Gadd DA et al. (2024), Nature Aging (https://doi.org/10.1038/s43587-024-00655-7)

• Smith A et al. (2025), Scientific Reports (https://doi.org/10.1038/s41598-025-06232-1)

Both use overlapping proteomic platforms and participant samples, and provide cause-specific prediction scores across 23–27 disease outcome. Both papers should be cited and briefly compared. Gadd et al. also reports death as an outcome. A critical discussion is required concerning the differing Area Under the Curve (AUC) values reported in the Gadd et al. paper (0.81 for the 10-year time frame for time to death) and in the current manuscript (AUC ≈ 0.66). The current manuscript includes more proteins which is a strength but it is the only one focused solely on mortality. The authors must more clearly articulate what new insight their analyses offer beyond these studies.

Model evaluation and replication

Panel derivation and testing are performed within the same dataset using a single 70/30 split. This limits generalizability and likely inflates performance metrics. The authors should implement stronger internal validation for instance bootstrapping or repeated cross-validation and discuss the absence of external replication. Any mortality risk score intended for potential future clinical use should demonstrate stable performance.

Minor points

-The abstract and title should be revised to avoid overstating predictive value. “Moderately improves” or similar wording would be more appropriate.

-The abstract should report actual AUC values.

Reviewer #2: Koziar et al. present a comprehensive analysis of plasma proteomic profiles of 38,150 UK Biobank participants, aiming to develop predictive models for all-cause mortality within 5 and 10 years of follow-up. Using robust statistical methods, they identified protein panels that modestly outperform traditional risk factors (with areas under receiver operating curves (AUCs) reaching up to approximately 0.68). Their study is methodologically and statistically sound and makes great use of a large data set but has several limitations affecting the potential clinical relevance of the findings.

Major points:

- The most significant limitation is the lack of an independent or external validation cohort. All analyses were conducted within the UK Biobank, which is a relatively homogenous population. This restricts the generalisability of the results (as the authors themselves acknowledge). I would recommend at least implementing an internal validation technique, e.g. cross-validation or bootstrapping, to better assess model robustness.

- The reported AUCs (all below 0.7) indicate limited predictive utility for clinical practice and appear modest compared to similar proteomic studies, which achieved AUCs of ~0.8 (e.g., Gadd et al., Nat Aging, 2024). Although the authors make note of this, a more detailed discussion of what factors might explain this relatively lower performance would be helpful (in particular, as to how such modest improvements over traditional risk factors might still translate into a clinical benefit).

- Using a broad “all-cause mortality” endpoint, excluding only accidental deaths, encompasses a wide range of conditions (e.g., cancer, cardiovascular disease, infections or autoimmune conditions). Each has potentially distinct proteomic signatures, and stratified analyses by major disease categories could identify more specific and biologically meaningful associations.

- The discussion would benefit from a more detailed mechanistic evaluation of the identified key proteins. While some biological context is provided, a deeper exploration of specific functions and pathways related to mortality risk would help to understand the biological plausibility of the findings.

Minor points

- Figure readability could be improved by increasing font sizes (particularly in multi-panel plots).

- The identified protein panel does not clearly suggest actionable targets for screening or therapeutic interventions. A more detailed review of their biological roles or the implications of those key proteins would strongly enhance the discussion.

While this is a valuable exploratory study in the field of proteomic biomarker discovery, the findings should be interpreted cautiously. The modest predictive gains and lack of an external validation cohort limit the immediate clinical translation. I recommend major revision, so that the authors can address the validation concerns and provide a more detailed assessment of clinical utility and biological relevance.

**Do you want your identity to be public for this peer review?** For information about this choice, including consent withdrawal, please see our Privacy Policy

Reviewer #1: No

Reviewer #2: No

---

## [Author Response · Author response to Decision Letter 1]

6 Oct 2025

Thank you for taking the time to review our manuscript and for your thoughtful comments. Please find the detailed responses below and the corresponding revisions/corrections in track changes in the re-submitted files.

Reviewer #1: Comments to the authors

The authors analyse >2,900 plasma proteins in a large prospective UK Biobank cohort (n = 38,150) and propose 6-and 10-protein panels for predicting all-cause mortality at 5 and 10 years (best AUC ≈ 0.66). While the dataset is highly valuable and the topic is of importance, the current version of the manuscript contains significant conceptual and methodological flaws that must be addressed before the results can be properly evaluated. The manuscript could make a valuable contribution if it is revised to address cause-of-death heterogeneity and position its findings within the context of recent large-scale proteomics work.

Author response: Thank you for seeing the value in this study, we have corrected the manuscript following the reviewers’ suggestions and believe this has improved the manuscript greatly.

Major points:

Outcome definition

The study treats all-cause mortality as a unified endpoint, however this aggregates biologically diverse causes of death like cardiovascular, cancer, respiratory, infection or neurological death. Without cause-specific breakdown or analyses, the current protein panel is difficult to interpret mechanistically or to apply clinically. For example, proteins associated with infection-related mortality may be diluted when pooled with proteins associated cancer deaths and vice versa. A table summarizing leading causes of death by ICD-10 code, with a breakdown in numbers and percentages and corresponding stratified hazard models is needed to demonstrate the true scope and relevance of the findings. Biomarker panels for mortality have limited utility if they cannot clarify why a patient is at risk. For clinical use risk stratification linked to potential actions like cardiovascular prevention or cancer screening is of relevance. The authors should justify the focus on undifferentiated all-cause-mortality or reframe their analysis around clinically distinct outcomes.

Author response: We thank the reviewer for their suggestions. We agree a table summarising the leading causes of death by ICD-10 codes would provide meaningful insight to our findings and have now included this in our paper (Table 2). We further carried out additional sensitivity analysis by running cox proportional hazards models for each of the two leading causes of mortality in our data (cardiovascular, and cancer – together accounting for the majority of deaths in our dataset) and a third category for all other disease-specific causes of death. The findings are now summarised in the manuscript (lines 147-168), with complete results available in the supplementary material (Supplementary Table 8).

As demonstrated in the results, the proteins present in our all-cause mortality panel have also demonstrated significant associations with disease-specific causes of death. While we agree with the reviewer’s points and that disease-specific analyses can provide mechanistic insights and potentially refine clinical translation, the focus of our study is to demonstrate that risk for all-cause mortality (regardless of specific cause – but excluding accidents for example) is reflected within the proteome (i.e. risk of death has an underlying biological signature – within this population of middle-aged and older individuals), not to directly provide a clinically usable biomarker panel. As the leading causes of death in this population are cancer and cardiovascular-related causes (covering the majority of cases), the specific signatures are not being “diluted” but are rather reflecting the primary drivers of the population mortality. Nevertheless, we have also reworded the manuscript throughout to ensure it reflects our focus more accurately.

Relationship to prior work

Recent UK Biobank studies have already examined protein-based disease prediction across multiple outcomes:

• Gadd DA et al. (2024), Nature Aging (https://doi.org/10.1038/s43587-024-00655-7)

• Smith A et al. (2025), Scientific Reports (https://doi.org/10.1038/s41598-025-06232-1)

Both use overlapping proteomic platforms and participant samples and provide cause-specific prediction scores across 23–27 disease outcome. Both papers should be cited and briefly compared. Gadd et al. also reports death as an outcome. A critical discussion is required concerning the differing Area Under the Curve (AUC) values reported in the Gadd et al. paper (0.81 for the 10-year time frame for time to death) and in the current manuscript (AUC ≈ 0.66). The current manuscript includes more proteins which is a strength, but it is the only one focused solely on mortality. The authors must more clearly articulate what new insight their analyses offer beyond these studies.

Author response: We thank the reviewer for their comment and drawing our attention to the two relevant studies. We have revised the manuscript to cite and discuss the mentioned papers, including the new insight our own research offers (lines 323-332).

In regards to the higher AUC reported in Gadd et al.; the number of features used in predicting death as an outcome was drastically higher and included over 200 proteins, putting their models at risk of being overfitted. In contrast, taking a parsimonious approach, our models have narrowed down the number of proteins to a handful, which are still demonstrating some predictive ability – highlighting that those few proteins can still pick up a biological signal for risk of death. If our 5-year all-cause mortality model was expanded to include 32 proteins, the AUC increases to ~0.7 (see Figure 3) and likely to increase further the more proteins are included. Additionally, Gadd et all, did not stratify into 5-year and 10-year mortality, and in effect, their analysis is only comparable to 10-year mortality risk. Further, Gadd et al. have not reported all performance metrics that would elucidate the real predictive ability of their model and help interpret how generalisable their models are. For example, they do not report the precision values which would help identify the rate of false positives, and while an AUC can be high, it could also have poor precision. We report the precision values for our models in Figure 4c.

Model evaluation and replication

Panel derivation and testing are performed within the same dataset using a single 70/30 split. This limits generalizability and likely inflates performance metrics. The authors should implement stronger internal validation for instance bootstrapping or repeated cross-validation and discuss the absence of external replication. Any mortality risk score intended for potential future clinical use should demonstrate stable performance.

Author response: We thank the reviewer for their suggestion. We have implemented internal validation for our models using k-fold cross-validation (see Methods, lines 429-430).

Minor points

-The abstract and title should be revised to avoid overstating predictive value. “Moderately improves” or similar wording would be more appropriate.

-The abstract should report actual AUC values.

Author response: Thank you, we have now changed the abstract and title as suggested.

Reviewer #2: Koziar et al. present a comprehensive analysis of plasma proteomic profiles of 38,150 UK Biobank participants, aiming to develop predictive models for all-cause mortality within 5 and 10 years of follow-up. Using robust statistical methods, they identified protein panels that modestly outperform traditional risk factors (with areas under receiver operating curves (AUCs) reaching up to approximately 0.68). Their study is methodologically and statistically sound and makes great use of a large data set but has several limitations affecting the potential clinical relevance of the findings.

Major points:

- The most significant limitation is the lack of an independent or external validation cohort. All analyses were conducted within the UK Biobank, which is a relatively homogenous population. This restricts the generalisability of the results (as the authors themselves acknowledge). I would recommend at least implementing an internal validation technique, e.g. cross-validation or bootstrapping, to better assess model robustness.

Author response: We thank the reviewer for their advice and have now implemented internal validation using k-fold cross-validation for our models (See Methods, lines 429-430). Unfortunately, the UK Biobank is a unique dataset and thus external validation is hard to achieve – we have addressed this point further within the discussion section:

“Due to the unique nature of the UK Biobank, our study presently lacks an external validation, limiting the generalisability of our findings; further, as a result of the vast majority of UK Biobank participants being of European ancestry, the predictive power of our protein panel is yet to be established for other ethnic groups.”

- The reported AUCs (all below 0.7) indicate limited predictive utility for clinical practice and appear modest compared to similar proteomic studies, which achieved AUCs of ~0.8 (e.g., Gadd et al., Nat Aging, 2024). Although the authors make note of this, a more detailed discussion of what factors might explain this relatively lower performance would be helpful (in particular, as to how such modest improvements over traditional risk factors might still translate into a clinical benefit).

Author response: We thank the reviewer for their comment. We have extended our discussion to include how modest improvements over traditional risk criteria can still be of value (lines 323-332). Regarding the reported AUCs of Gadd et al: the number of features used in predicting death as an outcome was drastically higher and included over 200 proteins, putting their models at risk of being overfitted. In contrast, our models have narrowed down the number of proteins to a handful, which are still demonstrating some predictive ability – highlighting that those few proteins can still pick up a biological signal for risk of death. Additionally, Gadd et al, did not stratify into 5-year and 10-year mortality, and in effect, their analysis is only comparable to 10-year mortality risk. Further, Gadd et al. have not reported all performance metrics that would elucidate the real predictive ability of their model and help interpret how generalisable their models are. For example, they do not report the precision values which would help identify the rate of false positives, and while an AUC can be high, it could also have poor precision. We report the precision values for our models in Figure 4c.

- Using a broad “all-cause mortality” endpoint, excluding only accidental deaths, encompasses a wide range of conditions (e.g., cancer, cardiovascular disease, infections or autoimmune conditions). Each has potentially distinct proteomic signatures, and stratified analyses by major disease categories could identify more specific and biologically meaningful associations.

Author response: We thank the reviewer for their suggestion. We have included a table summarising the leading causes of death in the dataset used for this study, by ICD10 code (Table 2 and Supplementary Table 3), as well as including sensitivity analysis in the form of additional cox proportional hazard models for disease-specific mortality, summarising the results in our manuscript (lines 147-168). As demonstrated in the results, the proteins present in our all-cause mortality panel have also demonstrated significant associations with disease-specific causes of death. The focus of our study is to demonstrate that risk for all-cause mortality (regardless of specific cause – but excluding accidents for example) is reflected within the proteome (i.e. risk of death has an underlying biological signature – within this population of middle-aged and older individuals), not to directly provide a clinically usable biomarker panel. As the leading causes of death in this population are cancer and cardiovascular-related causes (covering the majority of cases), the specific signatures are not being “diluted” but are rather reflecting the primary drivers of the population mortality. Therefore, we feel any further disease-specific mortality stratification is not required to demonstrate the central premise of our study. Nevertheless, we have also reworded the manuscript throughout to ensure it reflects our focus more accurately.

Minor points

- Figure readability could be improved by increasing font sizes (particularly in multi-panel plots).

Author response: Thank you, we have improved figure readability by displaying fewer elements on plots and increasing font size.

- The identified protein panel does not clearly suggest actionable targets for screening or therapeutic interventions. A more detailed review of their biological roles or the implications of those key proteins would strongly enhance the discussion.

- The discussion would benefit from a more detailed mechanistic evaluation of the identified key proteins. While some biological context is provided, a deeper exploration of specific functions and pathways related to mortality risk would help to understand the biological plausibility of the findings.

Author response: We thank the reviewer for their suggestion. We would like to clarify that the primary aim of our study was not to develop a clinical screening panel or to propose therapeutic targets, but rather to investigate the relationship between circulating proteins and all-cause mortality risk, as well as their predictive ability. While we agree that a detailed exploration of individual proteins could be valuable for future work, in-depth discussion of biological mechanisms falls outside the scope of our manuscript.

---

## [Decision Letter · Decision Letter 1]

19 Oct 2025

Dear Dr. Geifman,

Thank you for submitting your manuscript to PLOS ONE. After careful consideration, we feel that it has merit but does not fully meet PLOS ONE’s publication criteria as it currently stands. Therefore, we invite you to submit a revised version of the manuscript that addresses the points raised during the review process.

As you can see, both reviewers appreciated your work and only minor changes are needed at this stage.

We look forward to receiving your revised manuscript.

Kind regards,

Pavel Strnad

Academic Editor

PLOS ONE

Journal Requirements:

Reviewers' comments:

Reviewer's Responses to Questions

**Comments to the Author**

Reviewer #1: All comments have been addressed

Reviewer #2: All comments have been addressed

2. Is the manuscript technically sound, and do the data support the conclusions?

Reviewer #1: Yes

Reviewer #2: Yes

3. Has the statistical analysis been performed appropriately and rigorously?

Reviewer #1: Yes

Reviewer #2: Yes

4. Have the authors made all data underlying the findings in their manuscript fully available?

Reviewer #1: Yes

Reviewer #2: Yes

5. Is the manuscript presented in an intelligible fashion and written in standard English?

Reviewer #1: Yes

Reviewer #2: Yes

Reviewer #1: The authors addressed the critical points of reviewer 1 and 2. The authors now performed a cause specific analysis and provide the new data as Figure 2 and Suppl. Table 8 and Table 2 which strongly improves the manuscript.

The new cause specific analysis (Suppl. Table 8), however confirms heterogeneity: 312 proteins associate with cardiovascular mortality (median HR≈2.08; max 6.09) versus 28 for cancer (median HR≈1.59; max 2.79) and 139 for other causes. Yes, there is a shared core (19 proteins across all three causes, and many Cancer/Other proteins overlap with CVD), but the CVD set is far larger and contains many unique proteins, so pooling causes into “all cause death” inevitably averages/dilutes the stronger CVD specific signals. This is why the cause specific HRs are often higher than the corresponding all cause HRs.

Specifically, among the 5 year panel, SERPINA1 shows no cause specific association at Bonferroni; SERPINA3 is cancer specific; CRIM1/PLAUR are CVD/Other; only DDR1 and LTBP2 span all three causes. Similar patterns hold for the 10 year panel. Moreover, all cause HRs are consistently lower than the strongest cause specific HRs (e.g., CRIM1 6.09 for CVD vs 2.71 all cause), which I repeat indicates dilution when pooling etiologies. The data therefore support a partial shared signature of mortality risk, however not an etiology agnostic one; the manuscript should therefore adjust its wording accordingly and keep claims conservative given the modest all cause AUCs (~0.62–0.68).

The new cause specific table (Suppl. Table 8) is helpful as one can derive pairwise overlaps

o Share of Cancer proteins that are also in CVD: 23/28 = 82% (high)

o Share of CVD proteins that are also in Cancer: 23/312 = 7.4% (low)

The Cancer proteins set is small and mostly nested inside the much larger CVD set.

There is a substantial common core between CVD and Other, but CVD still has many unique proteins.

Revision required for minor points.

1. It is important that the headline claim is moderated in all instances. Please change everywhere “irrespective of etiology” to something like

“a small set of proteins captures a shared, non specific mortality risk signature, while effect sizes differ by cause.”

2. Add one sentence in Results summarizing the heterogeneity using the counts already in Suppl. Table 8 (e.g., “CVD 312 proteins, cancer 28, other 139; 19 proteins shared across all three”).

3. Acknowledge dilution explicitly: note that all cause HRs are lower than the strongest cause specific HRs for several panel proteins (e.g., CRIM1, PLAUR).

4. Data/code availability: include the repository link for scripts + derived, de identified outputs) to satisfy PLOS ONE policy.

Reviewer #2: The authors comprehensively addressed all points raised in the initial review. The revised abstract, manuscript, and figures enhance the clarity of their findings, and all findings are appropriately contextualised. The implementation of internal cross-validation and disease-specific analyses strengthens the manuscript's message.

**Do you want your identity to be public for this peer review?** For information about this choice, including consent withdrawal, please see our Privacy Policy

Reviewer #1: No

Reviewer #2: No

---

## [Author Response · Author response to Decision Letter 2]

29 Oct 2025

We thank you for the review of our manuscript and for your thoughtful comments. Please find the detailed responses below and the corresponding revisions/corrections in track changes in the re-submitted files.

Reviewer #1: The authors addressed the critical points of reviewer 1 and 2. The authors now performed a cause specific analysis and provide the new data as Figure 2 and Suppl. Table 8 and Table 2 which strongly improves the manuscript.

The new cause specific analysis (Suppl. Table 8), however confirms heterogeneity: 312 proteins associate with cardiovascular mortality (median HR≈2.08; max 6.09) versus 28 for cancer (median HR≈1.59; max 2.79) and 139 for other causes. Yes, there is a shared core (19 proteins across all three causes, and many Cancer/Other proteins overlap with CVD), but the CVD set is far larger and contains many unique proteins, so pooling causes into “all cause death” inevitably averages/dilutes the stronger CVD specific signals. This is why the cause specific HRs are often higher than the corresponding all cause HRs.

Specifically, among the 5-year panel, SERPINA1 shows no cause specific association at Bonferroni; SERPINA3 is cancer specific; CRIM1/PLAUR are CVD/Other; only DDR1 and LTBP2 span all three causes. Similar patterns hold for the 10-year panel. Moreover, all cause HRs are consistently lower than the strongest cause specific HRs (e.g., CRIM1 6.09 for CVD vs 2.71 all cause), which I repeat indicates dilution when pooling etiologies. The data therefore support a partial shared signature of mortality risk, however not an etiology agnostic one; the manuscript should therefore adjust its wording accordingly and keep claims conservative given the modest all cause AUCs (~0.62–0.68).

The new cause specific table (Suppl. Table 8) is helpful as one can derive pairwise overlaps

Share of Cancer proteins that are also in CVD: 23/28 = 82% (high)

Share of CVD proteins that are also in Cancer: 23/312 = 7.4% (low)

The Cancer proteins set is small and mostly nested inside the much larger CVD set.

There is a substantial common core between CVD and Other, but CVD still has many unique proteins.

Revision is required for minor points.

1. It is important that the headline claim is moderated in all instances. Please change everywhere “irrespective of etiology” to something like “a small set of proteins captures a shared, non-specific mortality risk signature, while effect sizes differ by cause.”

Author response: We thank the reviewer for their advice and have included the recommended changes to wording on lines 254 and 337-338.

2. Add one sentence in results summarising the heterogeneity using the counts already in Suppl. Table 8 (e.g., “CVD 312 proteins, cancer 28, other 139, 19 proteins shared across all three”).

Author response: We thank the reviewer for their suggestion and have included a summary of these results in the Abstract and in lines 157-158, 160, 162-163 and 165.

3. Acknowledge dilution explicitly, note that all cause HRs are lower than the strongest cause specific HRs for several panel proteins (e.g., CRIM1, PLAUR).

Author response: We thank the reviewer for their advice and have acknowledged the HR dilutions on lines 168-169.

4. Data code/availability include repository link for scrips + derived, de-identified outputs to satisfy PLOS ONE policy.

Author response: We thank the reviewer for this suggestion, the script used to obtain our results has been uploaded to https://github.com/natkoziar/A-Plasma-Based-Protein-Signature-Association-with-All-Cause-Mortality.git

However, due to the sensitive nature of the UK Biobank data, and restriction on access imposed by UK Biobank, we are unable to provide derived outputs beyond what has already been supplied in the Supplementary Data.

Reviewer #2: The authors comprehensively addressed all points raised in the initial review. The revised abstract, manuscript, and figures enhance the clarity of their findings, and all findings are appropriately contextualised. The implementation of internal cross-validation and disease-specific analyses strengthens the manuscript's message.

Author response: We thank the reviewer for their comment and are pleased to hear that the revisions have improved the clarity and contextualisation of our findings.

---

## [Editor Report · Decision Letter 2]

2 Nov 2025

A Plasma-Based Protein Signature Association with All-Cause Mortality

PONE-D-25-38384R2

Dear Dr. Geifman,

We’re pleased to inform you that your manuscript has been judged scientifically suitable for publication and will be formally accepted for publication once it meets all outstanding technical requirements.

Kind regards,

Pavel Strnad

Academic Editor

PLOS ONE

Additional Editor Comments (optional): Congratulations to the nice work!
---

## [Editor Report · Acceptance letter]

PONE-D-25-38384R2

PLOS ONE

Dear Dr. Geifman,

I'm pleased to inform you that your manuscript has been deemed suitable for publication in PLOS ONE. Congratulations! Your manuscript is now being handed over to our production team.

Kind regards,

on behalf of

Dr. Pavel Strnad

Academic Editor

PLOS ONE